# Gastric Infusion of Short-Chain Fatty Acids Improves Health via Enhance Liver and Intestinal Immune Response and Antioxidant Capacity in Goats

**DOI:** 10.3390/vetsci12050395

**Published:** 2025-04-22

**Authors:** Shaima Mohmed Nasr Abdu, Ismail Mohamed Abdalla, Yongkang Zhen, Chong Zhang, Zanna Xi, Jianjun Ma, Yuhong Zhong, Jiaqi Lin, Rahmat Ali, Mengzhi Wang

**Affiliations:** 1Laboratory of Metabolic Manipulation of Herbivorous Animal Nutrition, College of Animal Science and Technology, Yangzhou University, Yangzhou 225009, China; mohomedshema6@gmail.com (S.M.N.A.); yongkangzhenyzu@163.com (Y.Z.); 13899934184@163.com (C.Z.); xzn0552@163.com (Z.X.); mjj3833@163.com (J.M.); 18458390454@163.com (Y.Z.); 13328735568@163.com (J.L.); rahmatalihu@gmail.com (R.A.); 2College of Animal Science and Technology, Yangzhou University, Yangzhou 225009, China; ismailhmk@gmail.com

**Keywords:** antioxidant, gastrointestinal tract, immune, short-chain fatty acids

## Abstract

The gastrointestinal tract digests, absorbs, and metabolizes dietary nutrients and is considered a major immune organ in the body, with more than 70% of its immune cells. Short-chain fatty acids, including acetate, propionate, and butyrate, account for more than 70% of 75% of energy sources and can increase feed intake and body weight of ruminants as well as have anti-inflammatory, antioxidant, and antimicrobial effects. In this study, gastric infusion of short-chain fatty acids (acetate, propionate, and butyrate) in goats increased antioxidant enzyme activity, anti-inflammatory cytokines, and intestinal tight-junction protein expression. These results may contribute to improving goat intestinal health and productivity by boosting antioxidant capacity and modulating cytokines, and pro-inflammatory and anti-inflammatory cytokines.

## 1. Introduction

Over the last few decades, significant consideration has been given to improving intestinal health and preventing intestinal diseases in animals in order to increase their productivity as well as for economic development [1]. Therefore, the relationship between dietary intake, gut microbiota diversity, and function has been a research focus. There is increasing evidence implicating the gut microbiota and its product (short-chain fatty acids) as critical contributors to host health and gut/immune homeostasis [2]. Furthermore, a healthy digestive system of the host depends on the balanced crosstalk among intestinal microorganisms, epithelial barriers, and immune defenses in the gut mucosa, which are sustained by the activity of microorganisms producing bacterial metabolites such as short-chain fatty acids [3]. Short-chain fatty acids have garnered considerable interest among the various metabolites because of their health benefits [4], which are the dominant byproducts generated by anaerobic bacteria in the large intestine in the process of microbial breakdown of dietary fibers and protein. SCFAs refer to a group of fatty acids that have less than six carbon atoms. SCFAs are mainly composed of acetate, propionate, and butyrate, and more than 95% of the total SCFAs are accounted for by these three fatty acids [4]. In the intestinal contents, acetate, propionate, and butyrate are usually distributed 60:20:10 [5]. Fatty acids are necessary for growth and health, for immune system regulation and metabolism, as well as for improving intestinal epithelial barrier function [6]. Intestinal epithelial cells serve as the first physical barrier against microorganisms and participate in innate and adaptive immune responses [7]. Tight-junction proteins (TJPs) provide the physical barrier function, allowing nutrients from the diet to be absorbed while preventing molecules and pathogens from passing through the paracellular space [8]. TJPs have significant roles, e.g., the transmembrane proteins (claudins and occludins) help to maintain the integrity of the intestinal barrier for two adjacent epithelial cells [9]. Furthermore, zonula occludens-1 (ZO-1), a cytoplasmic adaptor protein, is linked to cytoskeletal tethering and transmembrane protein binding [10]. Recent cumulative in vitro investigations have indicated that SCFAs might modulate intestinal barrier integrity. For example, butyrate (15 mg/kg) has been shown to improve intestinal barrier integrity by promoting the relative mRNA expression of tight junction [11], while propionic acid inhibited the internalization of *Staphylococcus aureus* in bovine mammary epithelial cells [12]. On the other hand, SCFAs could downregulate the expression of pro-inflammatory cytokines in Caco-2 cells under lipopolysaccharide [13]. In pigs, in vivo studies have shown that dietary sodium butyrate supplementation (2 mM) may decrease *Clostridium* and *Escherichia coli* levels, increase Lactobacillus species, and maintain intestinal barrier function by reducing serum IL-6 and TNF-α levels [14]. Dietary supplementation with 1% SA, SP, or SB significantly enhanced crucian carp’s systemic antioxidant capacity, immune function, and microbial community structure [15].

The antioxidant capacity status can affect intestinal epithelial integrity, as the intestinal epithelium’s integrity is closely connected to the health of the gut [16]. Many factors in livestock production, such as environmental conditions and infections, can cause oxidative stress. This condition occurs when the levels of reactive oxygen species (ROS) and antioxidants, either intra- or extracellular, are not balanced, resulting in significant economic losses [17]. The body contains both enzymatic and non-enzymatic antioxidant systems, with the enzymatic antioxidant system consisting primarily of GSH-px and SOD [18]. One study found that sodium butyrate administration moderately increased the antioxidant capacity by increasing mRNA expression of various antioxidant genes, including Nrf2 (nuclear factor erythroid 2-related factor 2), SOD1, SOD2, and SOD3 [19].

Despite the extensive investigation of the immunomodulatory effects of SCFAs in mammals, previous research has primarily focused on disease treatment. However, the contribution of SCFAs in sustaining gut–immune interactions in healthy models has received less attention [15]. Moreover, most of the research is focused on butyrate. Other SCFAs, like acetate and propionate, are found in higher amounts in the gut and blood but have received less of the spotlight. Taking this into consideration, we hypothesized that gastric infusion of three SCFAs (acetate, propionate, and butyrate) would benefit goat intestinal health by increasing antioxidant capacity, pro-inflammatory and anti-inflammatory cytokines, and tight-junction proteins. To respond to this hypothesis, we evaluated the influence of infusing three different kinds of SCFAs, sodium acetate, sodium propionate, and sodium butyrate, on health performance and oxidative stress in a healthy goat model by investigating antioxidant enzyme activity in gastrointestinal tract organs, a physical and immune barrier, and tight-junction proteins.

## 2. Materials and Methods

### 2.1. Ethics Approval Statement

The Animal Care and Use Committee at Yangzhou University in Jiangsu, China (approval 202203-512), approved all ethical policies and procedures used in the animal experiments.

### 2.2. Goat Management and Experimental Design

Twenty adult Guanzhong milk goats 1.5 years of age, similar in body weight (47.44 + 3.38 kg), and in good health were selected for the experiment. Table 1 provides detailed information on the ingredients and chemical composition of the feed, based on dry matter. The ratio of concentrate to crude was 60:40. Feeding took place at 08:00 and 18:00 every day. Every goat was raised in a single pen with free access to water. Using single-factor complete randomization, the 20 dairy goats were randomly divided into 4 groups (*n* = 4/group). A gastric tube was used to perfuse orally equal volumes of 0.8 g/kg d sodium acetate, 0.8 g/kg d sodium propionate, and 0.8 g/kg d sodium butyrate solution. as the groups included an acetic acid group, a propionic acid group, and a butyric acid group, plus the control group without perfusion (the perfusion volume was 1 L, calibrated to be consistent with a pH meter before perfusion). After adaptive feeding, a gastric SCFA perfusion test was performed. The experiment lasted for 12 days, including 8 days for the pre-infusion period and 4 days for the formal infusion period.

### 2.3. Sampling Collections

All goats fasted for 12 h prior to the sampling day. Professional butchers followed the previous protocol for weighing, slaughtering, and dressing animals [20]. Shortly after the goats were slaughtered, the entire gastrointestinal tract was taken out, and the small intestine was separated from the rest of the digestive tract. There are three parts to the small intestine: the duodenum, which is defined by the Treitz ligament; the jejunum, which is defined by the ileocecal fold; and the ileum, which ends at the ileocecal junction after the mesentery has been carefully separated. We took an ileal epithelial cross-section tissue after the digesta was taken out. The tissue was then washed in phosphate-buffered saline solution (PBS), flash-frozen in liquid nitrogen, and kept at −80 °C so that we could use enzyme-linked immunosorbent assay (ELISA) methods to measure inflammatory cytokines and tight junctions.

### 2.4. Antioxidant Capacity Analysis

For antioxidant capacity analysis, we immediately removed the liver and intestine tissues and mixed them with an extraction solution while they were on ice, as detailed previously [20]. We extracted the supernatant by centrifuging the homogenized samples (8000 rpm for 10 min at 4 °C).

Measurement of protein concentrations was performed using a total protein quantitative assay kit (A045-2, Nanjing Jiancheng Bioengineering Institute, Nanjing, China). We measured the GSH-Px enzymatic activity using a colorimetric test kit (A005-1, Jiancheng). The activities of SOD (BC5165, Beijing Solarbio Science & Technology Co., Ltd., Beijing, China), T-AOC (BC1315, Solarbio), CAT (BC0205, Solarbio), GSH (BC1175, Solarbio), and MDA (BC0025, Solarbio) were examined using biochemical kits in accordance with the manufacturer’s protocols, with three replicates per measurement.

### 2.5. ELISA Analysis of Cytokines and Tight-Junction Proteins

For ELISA analysis, we immediately extracted the liver and intestine tissues and homogenized with extraction solution while they were on ice. See our previous work for more information [20]. We centrifuged homogenized samples at 2000–3000 rpm for 20 min at 4 °C to obtain the extract.

We used ELISA kits to check the concentration level of tight-junction proteins (claudin, occludin, and ZO-1) in the intestine, as well as the concentration level of proinflammatory and anti-inflammatory cytokines (TNF-α) and interleukin (IL-1, IL-6, and IL-10) in the liver and intestine. Measurements were performed according to the manufacturer’s instructions, with three replicates for each measurement.

### 2.6. Statistical Analysis

In our study, results were expressed as mean ± SD, with significance and high significance defined at *p* < 0.05 and *p* < 0.01. We applied one-way ANOVA or Kruskal–Wallis tests using SPSS 25.0 software (IBM-SPSS Inc., Chicago, IL, USA) to statistically compare the concentrations of SCFAs in gastrointestinal organs with the concentrations of inflammatory cytokines and tight junctions in ileal epithelial tissue. Graphics were produced using GraphPad Prism 6.0 software (GraphPad Software Inc., San Diego, CA, USA).

## 3. Results

### 3.1. Effects of SCFAs on Antioxidant Activity in the Liver and Intestine

The host’s health and homeostasis are closely related to its antioxidant capacity, which is an essential component of its defense system. The main antioxidant enzymes that safeguard the host from oxidative stress are T-AOC, SOD, CAT, GSH-Px, GSH, and MDA.

Compared with the control group, the activity of hepatic T-AOC was significantly downregulated in different groups (*p <* 0.05). In the intestine, the activity of T-AOC was significantly upregulated in the ileum and colon, with a non-significant difference between groups in the jejunum and cecum (*p <* 0.05), while goats in the SB group showed higher activity of T-AOC than other groups (Figure 1).

Compared with the control group, the activity of hepatic CAT was significantly upregulated in the SA group and downregulated in other groups (*p* < 0.05). Moreover, the activity of hepatic SOD was significantly upregulated in the SP group (*p* < 0.05), while there was no significant difference between the other groups (*p* < 0.05).

In the jejunum, the activity of CAT was significantly upregulated in the SP and SB group and was downregulated in the SA group (*p* < 0.05). Moreover, the activity of SOD in the jejunum was significantly downregulated in the SP group (*p* < 0.05); however, there was no significant difference between other groups. In the ileum, the activity of CAT was significantly downregulated in different groups (*p* < 0.05). There was no significant difference between the SP and SB groups. In addition, the activity of SOD in the ileum showed no significant difference between groups. In the colon, the activity of CAT was significantly upregulated in the SP and SB group (*p* < 0.05), while there was no significant difference between the control and SA groups (*p* < 0.05). Moreover, the activity of SOD in the colon was significantly upregulated in the SA and SB groups (*p* < 0.05), while there was no significant difference between the control and SP groups. In the cecum, the CAT activity showed no significant difference between groups. Moreover, the activity of SOD in the cecum was significantly downregulated in different groups (*p* < 0.05), but there was no significant difference between groups (Figure 2).

Compared with the control group, the activity of hepatic GSH-Px was significantly downregulated in different groups (*p* < 0.05). Moreover, hepatic GSH activity was significantly downregulated in the SA and SB groups (*p* < 0.05), whereas there was no significant difference between the control and SA groups. In the jejunum, the activity of GSH-Px was significantly downregulated in different groups (*p* < 0.05). Moreover, the activity of GSH in the jejunum was significantly downregulated in the SA and SP groups (*p* < 0.05). However, there was no discernible difference between the control and SB groups. In the ileum, the activity of GSH-Px was significantly downregulated in different groups (*p* < 0.05), with a non-significant difference between the SA and SB groups. Moreover, the activity of GSH in the ileum was significantly downregulated in the SA and SP groups (*p* < 0.05), while there was no significant different between the control and SB groups. In the colon, the activity of GSH-Px was significantly upregulated in the SA and SB groups (*p* < 0.05), while there was no significant difference between the control and SP groups. In addition, there was no significant difference between groups in the activity of GSH in the colon. In the cecum, the activity of GSH-Px showed no significant difference between groups. Furthermore, the activity of GSH in the cecum was significantly downregulated in different groups (*p* < 0.05) with a non-significant difference between them (Figure 3).

Lastly, the concentration of hepatic MDA was significantly upregulated in the SP group (*p* < 0.05), while there was no significant difference between the control and other groups. In the intestine, the concentration of MDA was significantly upregulated in the jejunum (*p* < 0.05) and downregulated in colon and cecum (*p* < 0.05), while there was a non-significant difference between groups in the ileum (Figure 4).

### 3.2. Effects of SCFAs on Cytokines Concentration in the Liver and Intestine

Compared to the control group, gastric infusion of SCFAs significantly decreased the level of hepatic IL-1 and IL-6 in different groups (*p* < 0.05), with no significant difference between them. Moreover, the level of IL-10 was significantly decreased in the SA and SB groups (*p* < 0.05), while there was a non-significant difference between the control and SP groups. Also, the level of TNF-α was significantly decreased in the SA group (*p* < 0.05), while there was a non-significant difference between the control and other groups (Figure 5).

The gastric infusion of SCFAs significantly increased the level of IL-1 in the jejunum between different groups in comparison to the control group (*p* < 0.05). Moreover, the level of IL-6 significantly decreased in the SB group (*p* < 0.05), with no significant difference between the control and other groups (*p* < 0.05). The level of IL-10 was significantly decreased in different groups (*p* < 0.05), with non-significant differences among them (*p* < 0.05), while there was no significant difference in the level of TNF-α between different groups (*p* < 0.05). In the ileum, the level of IL-1 was significantly decreased in the SB group (*p* < 0.05), with a non-significant difference between the control and other groups (*p* < 0.05). Also, the level of IL-10 was significantly decreased in different groups (*p* < 0.05), while there was no significant difference in the level of IL-6 and TNF-α between different groups (*p* < 0.05). In the colon, the level of IL-1 was significantly decreased in the SB group, with a non-significant difference between the control and other groups (*p* < 0.05). Besides, the level of IL-6 was significantly increased in the SA and SB groups (*p* < 0.05), while there was a non-significant difference between the control and the SA group (*p* < 0.05). In addition, the level of IL-10 was significantly increased in the SP group (*p* < 0.05), with a non-significant difference between the control and other groups (*p* < 0.05), while the level of TNF-α was significantly increased in different groups (*p* < 0.05), with no significant difference among them (*p* < 0.05). In the cecum, the levels of IL-6 and IL-10 were significantly increased in the SB group (*p* < 0.05), with a non-significant difference between the control and other groups (*p* < 0.05), while there was a non-significant difference in the levels of IL-1 and TNF-α between different groups (*p* < 0.05) (Figure 6).

### 3.3. Effects of SCFAs on Intestinal Tight Junction Proteins

Compared with the control group, gastric infusion of SCFAs showed non-significant difference in the level of claudin between different groups in the jejunum (*p* < 0.05). Moreover, the level of occludin was significantly increased in the SP and SB groups (*p* < 0.05), with no significant difference between them, while there was a non-significant difference between the control and the SA group. Furthermore, the level of ZO-1 was significantly decreased in the SA and SB groups (*p* < 0.05), with no significant difference between them, while there was a non-significant difference between the control and the SP group. In the ileum, the level of claudin was significantly increased in different groups (*p* < 0.05). Moreover, the level of occludin was significantly decreased in different groups (*p* < 0.05), while there was a non-significant difference in the level of ZO-1 in different groups (*p* < 0.05). In the colon, the level of claudin was significantly decreased in different groups (*p* < 0.05), with no significant difference between them (*p* < 0.05). Moreover, the level of occludin was significantly increased in the SA group (*p* < 0.05) and decreased in the SB group (*p* < 0.05), while there was a non-significant difference between the control and the SP group (*p* < 0.05). In addition, the level of ZO-1 was significantly decreased in the SA group. In the cecum, there was a non-significant difference in the level of claudin between different groups. Moreover, the level of occludin was significantly decreased in the SB group (*p* < 0.05), while there was a non-significant difference between the control and other groups. In addition, the level of ZO-1 was significantly decreased in the SP group and increased in the SB group (*p* < 0.05), while a non-significant difference was found between the control and the SA group (Figure 7).

### 3.4. Relationship Among Intestinal Antioxidant, Inflammatory Cytokine, and Tight-Junction Protein Indexes

This study examined the potential correlation among antioxidant, inflammatory cytokine, and tight-junction protein indexes in the liver and intestine. In the liver (Figure 8A), GSH-PX had a positive relationship with IL-6 (r = 0.892), while SOD had a negative relationship with IL-6 (r = −0.781). Moreover, the T-AOC and CAT were negatively correlated with TNF-α (r = −0.634 and −0.886, respectively), and T-AOC, GSH, CAT, and MDA were negatively correlated with IL-10 (r = −0.995, −0.883, −0.624, and −0.972, respectively). In the jejunum (Figure 8B), T-AOC had a highly positive relationship with ZO-1 (r = 0.919), IL-6 (r = 0.704), and IL-10 (r = 0.794), while it was negatively correlated with claudin (r = −0.998) and TNF-α (r = −0.863). However, GSH-Px was positively correlated with IL-6 (r = 0.711) and IL-10 (r = 0.896) and negatively correlated with IL-1β (r = −0.722), TNF-α (r = −0.628), and occludin (r = −0.62). GSH was negatively correlated with IL-1β (r = −0.929) and positively correlated with claudin (r = 0.534). CAT was positively correlated with occludin (r = 0.73) and claudin (r = 0.67) and negatively correlated with IL6 (r = −0.972) and IL-10 (r = −0.664). Moreover, SOD was negatively correlated with IL-1β (r = −0.965), whereas MDA was positively correlated with IL-1β (r = 0.633). In the ileum (Figure 8C), T-AOC had a negative relationship with IL-1β (r = −0.978), IL-6 (r = −0.611), and occludin (r = −0.847), while having a positive relationship with ZO-1 (r = 0.937). GSH-PX was positively correlated with IL-10 (r = 0.946) and claudin (r = 0.608). Additionally, GSH had a positive relation with IL-6 (r = 0.622) and claudin (r = 0.908). CAT was positively correlated with IL-1β (r = 0.727), IL-6 (r = 0.677), and occludin (r = 0.67). SOD had a positive relation with both IL-1β (r = 0.878) and occludin (r = 0.848), while it was negatively related with ZO-1 (r = −0.849). MDA was positively related with IL-6 (r = 0.943) and TNF-α (r = 0.931), while it had a negative relation with IL-10 (r = −0.721). In the colon (Figure 8D), T-AOC was positively related with ZO-1 (r = 0.665) and negatively with IL-1β (r = −0.999). In addition, the GSH-PX was positively correlated with IL-6 (r = 0.939) and TNF-α (r = 0.675), whereas it was inversely correlated with IL-1β (r = −0.75). GSH was highly positively correlated with claudin (r = 0.955) and negatively correlated with IL-10 (r = −0.81). CAT was highly positively correlated with IL-10 (r = 0.947) and SOD was positively correlated with IL-6 (r = 0.936) and TNF-α (r = 0.807), but inversely correlated with IL-1β (r = −0.894). MDA was positively correlated with IL-6 (r = 0.759) and negatively correlated with IL-10 (r = −0.972). In the cecum (Figure 8E), T-AOC had a positive relationship with ZO-1 (r = 0.841) and IL-6 (r = 0.682), but a negative relationship with IL-1β (r = −0.827) and occludin (r = −0.844), In addition, GSH-PX had a negative relationship with IL-1β (r = −0.87) and claudin (r = −0.924; GSH had a negative relationship with IL-1β (r = 0.678), whereas CAT was positively correlated with TNF-α (r = 0.661) and ZO-1 (r = 0.784). On the other hand, SOD had a negative relationship with IL-1β (r = −0.859) and claudin (r = −0.759), and MDA was inversely correlated with IL-1β (r = −0.75).

## 4. Discussion

In normal physiological conditions, reactive oxygen species (ROS) can be generated by the digestive tract [21]. Several factors, including infection, weaning, and environmental influences, can lead to oxidative stress. This condition occurs when the levels of ROS and antioxidants inside or outside of cells are out of balance [22]. As a result, livestock production is adversely affected [23]. The antioxidative enzymes T-AOC, SOD, CAT, GSHPX, GSH, and MDA play an essential role in the defense against oxidative stress in animals, including goats [24]. As natural antioxidant enzymes, GSH-Px, SOD, and CAT are important in the removal of endogenous free radicals generated by the host body [25]. In addition to maintaining the body’s oxidative balance, these enzymes play an essential role in its antioxidative and oxidative defense mechanisms [26]. As part of this study, we examined how SA, SB, and SP could modulate the activities of those antioxidative enzymes as well as the levels of MDA, a marker of oxidative stress, in goat liver and intestinal tracts. In previous research, SCFAs in the basal diet improved gut function and innate immunity responses in crucian carp. The four diet treatments were administered for six weeks: a control basal diet, a basal diet supplemented with 1% sodium acetate (BDSA), a basal diet supplemented with 1% sodium propionate (BDSP), and a basal diet supplemented with 1% sodium butyrate (BDSB). Based on the study results, it was concluded that dietary BDSP, BDSA, and BDSB may enhance host oxidative defense mechanisms by increasing the activity of antioxidative enzymes, including T-SOD, GSH-Px, and CAT [15]. In another research study, dietary oregano essential oil supplementation significantly increased SOD, CAT, GSH-Px, and T-AOC activities and decreased MDA levels [27].

Several investigations have demonstrated that butyrate has significant antioxidative effects, both in vivo and in vitro [28]. Our study demonstrated that the SB group significantly upregulated T-AOC activity in the ileum, CAT activity in the jejunum, and T-AOC, CAT, GSH-Px, and SOD activity in the colon. An in vitro study has shown that butyrate could upregulate GPx-3, GPx-4, and total GSH-Px mRNA expressions in vascular smooth muscle cells [29]. The GSH-Px activity increased linearly with increasing SB levels in liquid feeds (milk and/or milk replacer) for pre-weaned dairy calves [30]. The serum MDA concentration decreased linearly as SB levels were increased, suggesting that the SB supplementation may have boosted the oxidative stress defense system while attenuating oxidative stress [31]. A prior investigation found that dietary SB increased SOD activity and decreased serum MDA concentration in chickens [32]. In the trial, to assess the impact of supplemented dietary-coated sodium butyrate (CSB) levels on intestinal antioxidants, immune function, and cecal microbial populations in laying hens, the results indicated that CSB supplementation not only increased SOD activity in the jejunum but also increased T-AOC activity in the ileum. According to the results, CSB inclusion may boost the antioxidant defense system through enzymatic and nonenzymatic mechanisms. In addition, they observed a decrease in MDA levels in the jejunum of laying hens after they were treated with CSB, indicating that CSB might avoid lipid peroxidation in the intestinal tract. The integrated results confirmed that CSB has a beneficial effect on the intestinal antioxidant system of laying hens [33]. In the present study, CAT activity was significantly upregulated in the intestine (jejunum and colon) in the SP group, and the SP group also significantly upregulated MDA activity in the jejunum. This is in agreement with the study investigating the effects of different doses (0%, 0.5%, 1%, and 2%) of sodium propionate on antioxidant capacity in common carp, which showed that dietary SP significantly upregulated antioxidant enzyme genes [34]. In contrast to these findings, yellow catfish fed SCFA blends displayed decreased CAT and SOD enzyme activities [35]. According to Safari et al. [36], zebrafish fed dietary SP showed downregulation of antioxidant enzyme genes (SOD, CAT).

The intestinal tract, the body’s largest immune organ, protects pathogens from penetrating host tissues via operational immunological protection and serves as a physical barrier [37]. In innate immune defense, cytokines such as IL-1β, IL-6, TNF-α, and IL-10 play an important role [38]. After induction, they stimulate the immune system’s resistance to disease and intensify its immunological reaction. In this study, basal diets supplemented with SA, SB, and SP were shown to up-regulate IL-1β expression in the jejunum between different groups compared to the control group. IL-6 and IL-10 levels in the colon were significantly increased in the SA and SB groups, while TNF-α levels were significantly increased in each group. In line with our research, dietary sodium propionate could increase the expression of pro-inflammatory cytokines (IL-1β and TNF-α) in healthy zebrafish [39]. In a study by Reda et al., propionic acid/salt and a formic mixture were reported to increase the expression of TNF-α and IL-1β [40]. It was found by Liu et al. that using a certain type of microencapsulated sodium butyrate (MSB 3.0) could increase the levels of both TGF-β and TNF-α in the hindgut of common carp [41]. In a study by Standen et al., adding a commercial mixed-species probiotic to the diet could raise levels of both pro-inflammatory (TNF-α and IL-1β) and anti-inflammatory cytokines (TGF-β and IL-10) [42]. The author also thought that the anti-inflammatory effects of TGF-β and IL-10 might be linked to the increase in pro-inflammatory expression. According to previously published literature, butyrate exerts a powerful effect on modulating immune systems through stimulation of inflammatory cytokines [43]. In our study, the levels of hepatic IL-1, IL-6, and IL-10 were significantly decreased in the SB group compared to the control group, while no significant differences were observed in hepatic TNF-α. In the intestine, the SB group downregulated the level of IL-6 and IL-10 in the jejunum, IL-1 and IL-10 in the ileum, and IL-1 in the colon, while the SB group upregulated the level of IL-6 and TNF in the colon and the level of IL-6 and IL-10 in the cecum. In the study of CSB, a similar effect was exerted by the dietary treatment of CSB. The observed effect might have been the downregulation of the expression of pro-inflammatory cytokines (TNF and IL-6), concomitant with increased expression of anti-inflammatory cytokines (IL-10) in the intestinal cells [33]. TNF and IL-6 are considered to contribute to gut damage and are abundant in the inflamed gut [44]. Numerous studies have demonstrated the beneficial effects of IL-10 on intestinal permeability [45]. As well as promoting intestinal epithelial cell proliferation, IL-10 is also known to help maintain the intestinal mechanical barrier [46]. In a study conducted in a pig model, sodium butyrate supplementation resulted in a reduction in pro-inflammatory cytokines (IL-6 and TNF-α) in the serum [47]. In an in vitro study, we revealed that SCFAs (1 to 5 mM butyrate, 1 to 5 mM propionate, or 20 mM acetate) effectively suppressed IL-8 and IL-1β mRNA abundances in Caco-2 cells induced by LPS [29]. Similarly, in our study, gastric infusion of SCFAs decreased IL-1β levels in the ileum and colon.

Claudin, occludin, and ZO-1 are the major tight junction proteins and play a key role in enhancing the physical barrier of the intestinal tract. Their higher gene expression levels are associated with barrier function [48]. The present study demonstrated that gastric infusion of SCFAs significantly increased levels of occludin in SP and SB in the jejunum, significantly increased levels of claudin among the three groups in the ileum, significantly increased levels of occludin in the SA and SP groups as well as ZO-1 in the SB group in the colon, and significantly increased levels of ZO-1 in the cecum. In a previous study, the BDSA, BDSP, and BDSB groups had significantly higher levels of gene expression for ZO-1 and occludin than the BD group. This suggests that short-chain fatty acids in the diet may help the gut barrier work better [15]. Furthermore, dietary sodium butyrate at a concentration of 0.2% has been found to increase the expression of intestinal gene profiles relating to occludin, ZO-1, and occludin 4 in juvenile turbot [49]. Feng et al. reported that SB significantly increased the expression of ZO-1 and occludin in the intestinal tract both in vitro and in vivo [50]. There was a previous study that showed that different treatments of fatty acids improved the integrity and function of the intestinal epithelium [51]. In the study of different combinations of dietary supplementation with fatty acids, including sodium butyrate (SB), omega-3 polyunsaturated fatty acids (n-3 PUFAs), and medium-chain fatty acids (MCFAs), they observed that piglets in the control diet with coated SB (1 g/kg) and coated MCFAs (7.75 g/kg) showed significantly higher levels of mRNA expressions of claudin-1 and ZO-1 in the jejunal mucosa, thereby alleviating diarrhea in the piglets. Compared to the piglets in the control diet with coated SB (1 g/kg), coated MCFAs (7.75 g/kg), and coated n-3 PUFAs (68.2 g/kg), the piglets in the coated n-3 PUFA diet demonstrated higher expressions of claudin1, ZO-1, and occludin, suggesting that these three fatty acids may work synergistically to strengthen intestinal barrier function [52].

## 5. Conclusions

In conclusion, the results showed that SCFAs might safeguard the host against oxidative stress by increasing the activity of specific antioxidant enzymes (SOD, CAT, MDA, GSH, GSH-PX, and T-AOC) in the GIT of goats. Moreover, SCFA infusion demonstrated the potential to have certain positive effects on inflammatory cytokines (IL-1β, IL6, TNF-α, and IL-10) and tight-junction protein concentrations (claudin, occludin, and ZO-1) in the liver and intestine. Our findings revealed that the gastric infusion of SCFAs (SA, SP, and SB) have positive roles on GIT health and innate immunity in Guangzhong goats. In the future, we recommend using different dosages of SCFA additives in goat diets in order to obtain more positive results.

## Figures and Tables

**Figure 1 vetsci-12-00395-f001:**
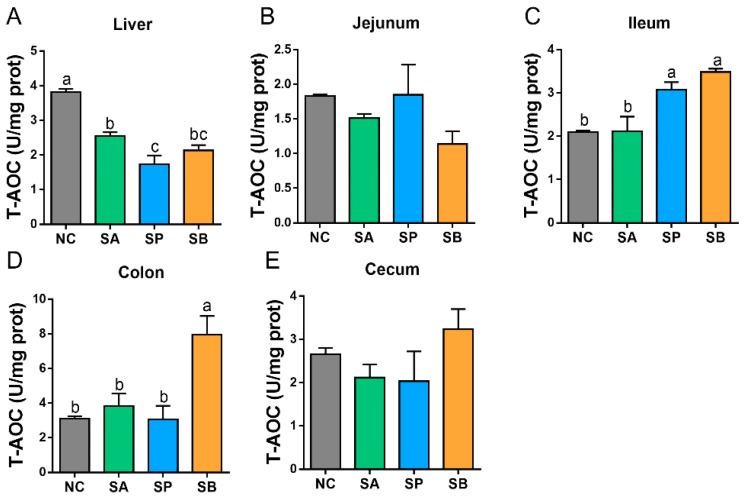
The effects of gastric infusion of SCFAs on T-AOC activities in the liver (**A**) and intestine (jejunum (**B**), ileum (**C**), colon (**D**), and cecum (**E**)) of goats. Different letters above the bars represent different groups that showed significant differences (*p* < 0.05); values are presented as the mean ± SE. Note: NC, control group; SA, sodium acetate; sodium propionate; sodium butyrate; T-AOC, total antioxidant capacity.

**Figure 2 vetsci-12-00395-f002:**
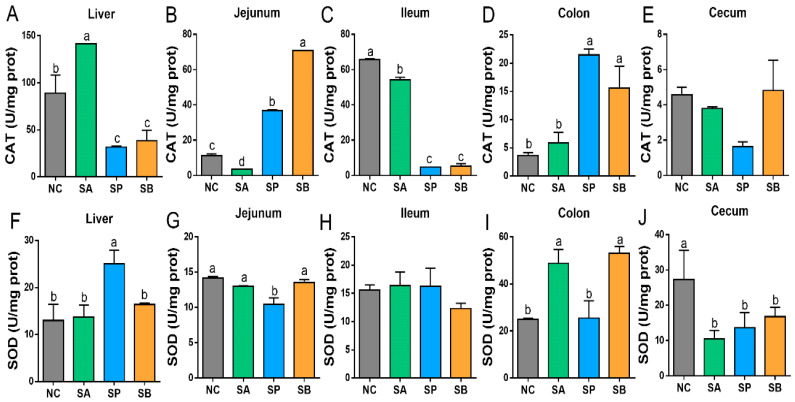
The effects of gastric infusion of SCFAs on CAT and SOD activities in the liver and intestine of goats. (**A**–**E**): Representative charts of the effects of gastric infusion of SCFAs on CAT in the liver, jejunum, ileum, colon, and cecum, respectively. (**F**–**J**): Representative charts of the effects of gastric infusion of SCFAs on SOD in the liver, jejunum, ileum, colon, and cecum, respectively. Different letters above bars represent different groups that showed significant differences (*p* < 0.05); values are presented as the mean ± SE. Note: NC, control group; SA, sodium acetate; sodium propionate; sodium butyrate; CAT, catalase; SOD, superoxide dismutase.

**Figure 3 vetsci-12-00395-f003:**
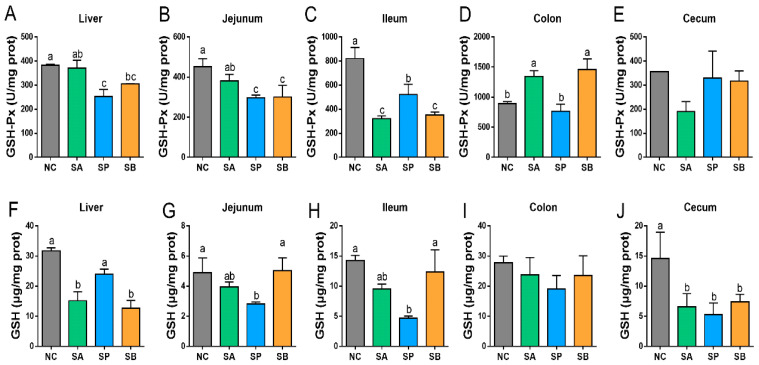
The effects of gastric infusion of SCFAs on GSH-Px and GSH activities in the liver and intestine of goats. (**A**–**E**): Representative charts of the effects of gastric infusion of SCFAs on GSH-Px in the liver, jejunum, ileum, colon, and cecum, respectively. (**F**–**J**): Representative charts of the effects of gastric infusion of SCFAs on GSH in the liver, jejunum, ileum, colon, and cecum, respectively. Different letters above bars represent different groups that showed significant differences (*p* < 0.05); values are presented as the mean ± SE. Note: NC, control group; SA, sodium acetate; sodium propionate; sodium butyrate; GSH-Px, glutathione peroxidase; GSH, glutathione.

**Figure 4 vetsci-12-00395-f004:**
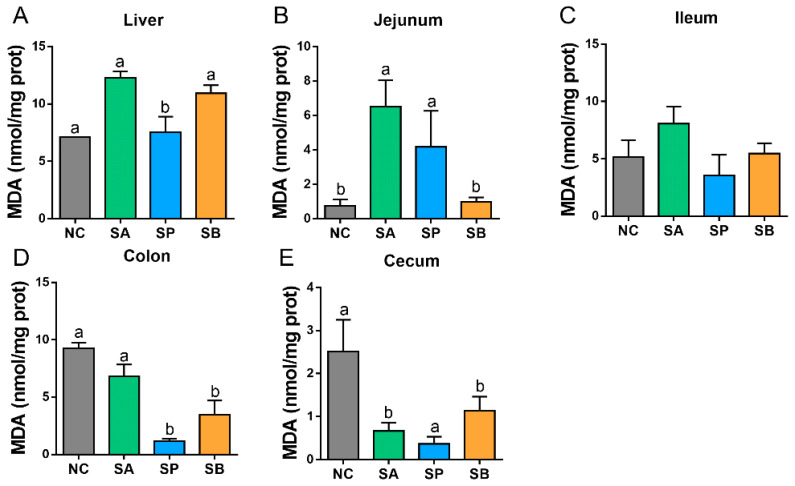
The effects of gastric infusion of SCFAs on MDA activities in the liver (**A**) and intestine (jejunum (**B**), ileum (**C**), colon (**D**), and cecum (**E**)) of goats. Different letters above bars represent different groups that showed significant differences (*p* < 0.05); values are presented as the mean ± SE. Note: NC, control group; SA, sodium acetate; sodium propionate; sodium butyrate; MDA, maleic dialdehyde.

**Figure 5 vetsci-12-00395-f005:**
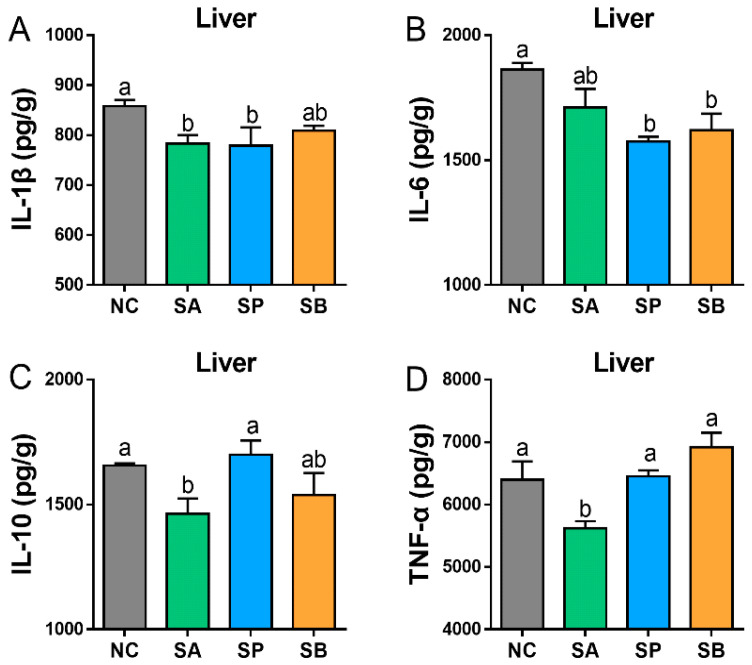
Effect of gastric infusion of SCFAs on cytokine concentration (IL-1β (**A**), IL-6 (**B**), IL-10 (**C**), and TNF-α (**D**)) in the liver of goats. Different letters above bars represent different groups that showed significant differences (*p* < 0.05); values are presented as the mean ± SE. Note: NC, control group; SA, sodium acetate; sodium propionate; sodium butyrate; IL-1β, interleukin-1 beta; IL-6, interleukin-6; IL-10, interleukin-10; TNF-α, tumor necrosis factor alpha.

**Figure 6 vetsci-12-00395-f006:**
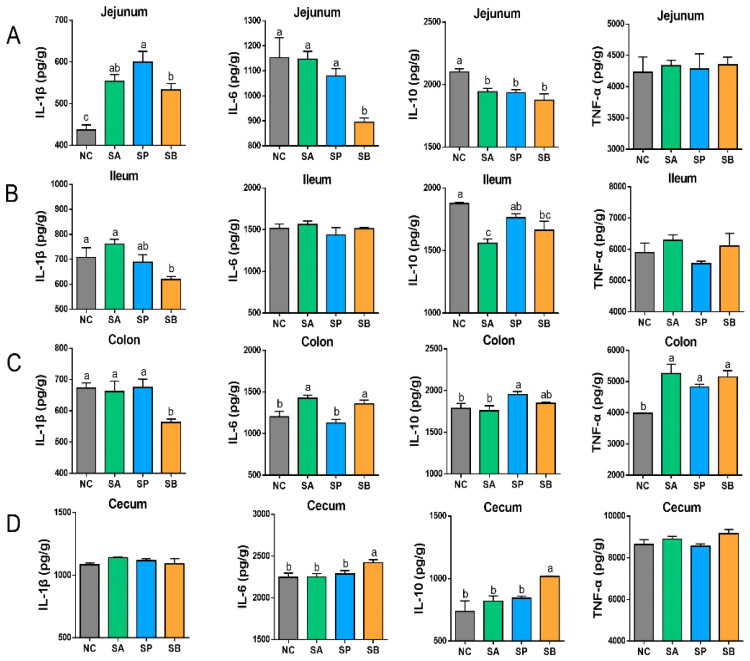
Effect of gastric infusion of SCFAs on cytokine concentration in the jejunum (**A**), ileum (**B**), colon (**C**), and cecum (**D**) of goats. Different letters above bars represent different groups that showed significant differences (*p* < 0.05); values are presented as the mean ± SE. Note: NC, control group; SA, sodium acetate; sodium propionate; sodium butyrate; IL-1β, interleukin-1 beta; IL-6, interleukin-6; IL-10, interleukin-10; TNF-α, tumor necrosis factor alpha.

**Figure 7 vetsci-12-00395-f007:**
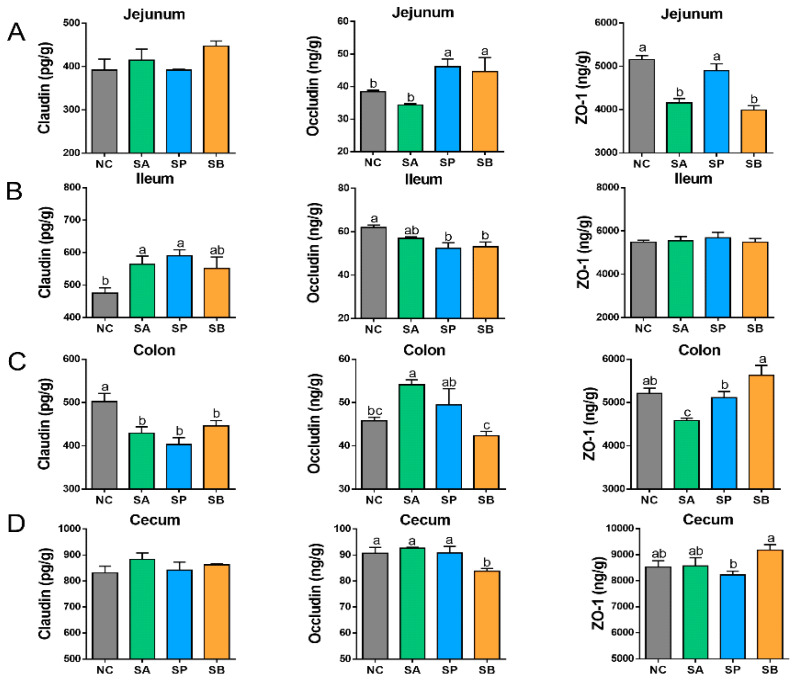
Effect of gastric infusion of SCFAs on the intestinal tight-junction proteins in the jejunum (**A**), ileum (**B**), cecum (**C**), and colon (**D**) of goats. Different letters above bars represent different groups that showed significant differences (*p* < 0.05); values are presented as the mean ± SE. Note: NC, control group; SA, sodium acetate; sodium propionate; sodium butyrate.

**Figure 8 vetsci-12-00395-f008:**
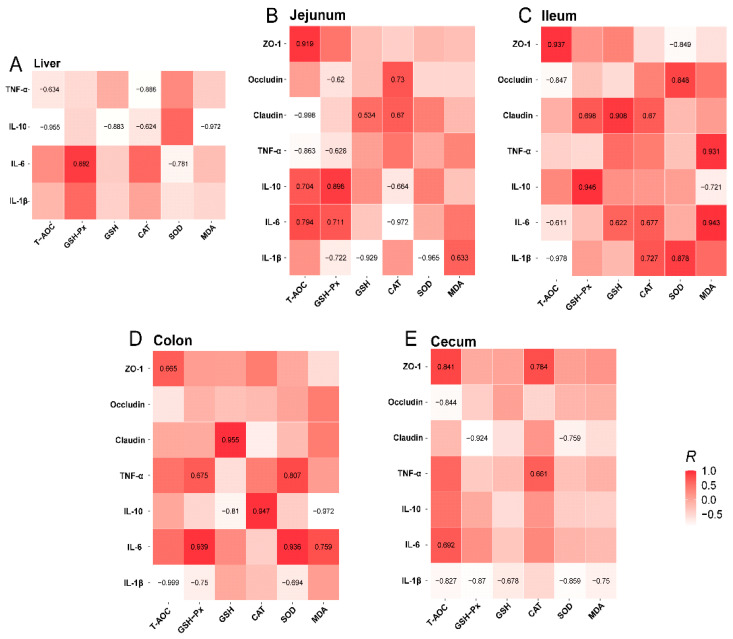
Relationship among intestinal antioxidant, inflammatory cytokine, and tight-junction protein indexes in the liver (**A**), jejunum (**B**), ileum (**C**), colon (**D**), and cecum (**E**). Note: IL-1β, interleukin-1 beta; IL-6, interleukin-6; IL-10, interleukin-10; TNF-α, tumor necrosis factor alpha; ZO-1, zonula occludens-1; T-AOC, total antioxidant capacity; CAT, catalase; SOD, superoxide dismutase; GSH-Px, glutathione peroxidase; GSH, glutathione; MDA, maleic dialdehyde.

**Table 1 vetsci-12-00395-t001:** Experimental diet formula and nutritional value (dry matter basis).

Ingredients	% of DM	Nutrients	g/kg of DM
Oat hay	41.00	DE (MJ/kg)	10.25
Corn	29.50	CP	155.60
Soybean meal	14.50	NDF	352.60
Wheat bran	10.00	ADF	196.25
Stone dust	0.35	EE	28.94
CaH_2_PO_4_	0.15	Ca	4.08
NaCl	0.50	P	4.65
Premix	4.00		
Total	100.00		

Note: DM, dry matter; DE, digestible energy; CP, crude protein; NDF, neutral detergent fiber; ADF, acid detergent fiber; EE, ether extract. The following premix was provided per kg of diet: VA 200,000 IU, VD3 70,000 IU, VE 350 IU, Fe 1.6 g, Cu 1.7 g, Zn 8.2 g, Mn 2.5 g, and Se 40 mg.

## Data Availability

All data generated or analyzed during this study are included in this manuscript and its information files.

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
