# Peer review of "Gastric Infusion of Short-Chain Fatty Acids Improves Health via Enhance Liver and Intestinal Immune Response and Antioxidant Capacity in Goats"

_vetsci, 2025, doi:10.3390/vetsci12050395_

Round 1

Reviewer 1 Report

Comments and Suggestions for Authors

Authors have conducted an interesting study. My decision is based on a limited information in the M&M section.

i.e. More details are required to procedures about infusions. Title indicates gastric and M&M section indicates rumen. Authors need to clarify where the infusion were applied and how was it verified.  There is no information how tissues were sampled. Were the goats euthanized ? In general, M&M section is limited to evaluate the procedures quality.

The main concern is about the statistical analysis of data. Authors indicated that “One‐way ANOVA followed by Duncan‘s multiple comparison test was used to
compare significant differences between multiple sets of data with p‐value < 0.05 (SPSS software). My suggestion is that conception of design and model should be reconsidered. The use of a MIXED model with random and fixed components is the right way.

Author Response

Comments and Suggestions for Authors

Authors have conducted an interesting study. My decision is based on a limited information in the M&M section.

Point 1: i.e. More details are required to procedures about infusions. Title indicates gastric and M&M section indicates rumen. Authors need to clarify where the infusion were applied and how was it verified.  There is no information how tissues were sampled. Were the goats euthanized? In general, M&M section is limited to evaluate the procedures quality.

Response 1: Thanks for your comment, we have added more details in M&M (sampling collection part), in lines 140-151.

Point 2: The main concern is about the statistical analysis of data. Authors indicated that “One‐way ANOVA followed by Duncan‘s multiple comparison test was used to compare significant differences between multiple sets of data with p‐value < 0.05 (SPSS software). My suggestion is that conception of design and model should be reconsidered. The use of a MIXED model with random and fixed components is the right way.

Respond 2: Thanks for your comment, we have corrected the statistical analysis description, in lines 175-181.

Reviewer 2 Report

Comments and Suggestions for Authors
  1. The abstract needs improvement on several points, such as, did the authors use significant words without mentioning any p-values? In addition, several abbreviations exist without any identification of the full name; also, authors should not start the sentence with abbreviations. The conclusion in the abstract needs to be modified according to the results, no data about the dose , animals and experimental period was mentioned  please
  2. Keywords should be rearranged according to the alphabetical orders 
  3. The introduction lacks clear hypothesis , in vitro  or in vivio should be italic ,
  4. Twenty adult Guanzhong milk goats of 1.5 years of age, similar in body shape (47.44  125

    +3.38kg).  body weight not shape also need to mentioned which period of lactation were  the animals

  5.  

    VFAs or SCFAs Please make it just one abbreviation
  6. Line 138 Table 1. This is a table. Tables should be placed in the main text near to the first time they are cited?
  7. Percent level  should be removed table 1
  8. , a rumen VFA perfusion test was performed.? how did you do that , which methodology ? what about doses? 
  9. Hepatic and intestinal tissues were immediately extracted and homogenized with - what about the slaughter ? 
  10. All Tables were repeated again? why the manuscript needs to Reorganized 
  11. 2.5. Statistical Analysis   need to reconsider again 
  12. The discussion so long and must be related to authors findings please 
Comments on the Quality of English Language

The English could be improved to more clearly express the research.

Author Response

Point 1: The abstract needs improvement on several points, such as, did the authors use significant words without mentioning any p-values? In addition, several abbreviations exist without any identification of the full name; also, authors should not start the sentence with abbreviations. The conclusion in the abstract needs to be modified according to the results, no data about the dose, animals and experimental period was mentioned please

Respond 1: Thanks for your comment, we have modified the abstract

Point 2: Keywords should be rearranged according to the alphabetical orders 

Respond: Thanks for your comment, Keywords have been rearranged according to the alphabetical orders The introduction lacks clear hypothesis, in vitro or in vivio should be italic,

Respond: Thanks for your comment, we have added clear hypothesis and in vitro or in vivio have been italic

Point: Twenty adult Guanzhong milk goats of 1.5 years of age, similar in body shape (47. 44. 125

+3.38kg).  body weight not shape also need to mentioned which period of lactation were the animals

Respond: Thanks for your comment, we have corrected this mistake

Point: VFAs or SCFAs Please make it just one abbreviation

Respond: Thanks for your comment, we have modified to SCFAs

Point: Line138 Table 1. This is a table. Tables should be placed in the main text near to the first time they are cited?

Respond: Thanks for your comment, we have corrected this mistake

Point: Percent level should be removed table 1

Respond: Thanks for your comment, percent level has been removed table 1

Point, a rumen VFA perfusion test was performed.? how did you do that, which methodology? what about doses? 

Response: Thanks for your comment, we have added more details in M&M (sampling collection part).

Point: Hepatic and intestinal tissues were immediately extracted and homogenized with - what about the slaughter? 

Response: Thanks for your comment, we have added more details in M&M (sampling collection part)

Point: All Tables were repeated again? why the manuscript needs to Reorganized 

Respond: Thanks for your comment, we have corrected this mistake

Point: 2.5. Statistical Analysis   need to reconsider again 

Respond: Thanks for your comment, we have modified the statistical analysis description

Point; The discussion so long and must be related to authors findings please 

Respond: Thanks for your comment, we have removed paragraph  

Comments on the Quality of English Language

The English could be improved to more clearly express the research.

Reviewer 3 Report

Comments and Suggestions for Authors

Comments and Suggestions for Authors

After reviewing the manuscript entitled “Gastric Infusion of Short Chain Fatty Acids Improves Healthvia Enhance Liver and Intestinal Immune Response and Antioxidant Capacity in Goat”, the following suggestions were made it. The manuscript contains interesting information on the in vivo effects of short-chain fatty acids on the antioxidant and immune capacity of goat liver and intestinal tissue. However, major corrections are required before the manuscript can be accepted for publication. The individual corrections are shown below:

Simple Summary

Lines 16-24: The authors used an excessive amount of background information, which could confuse the reader. Please shorten these lines by at least 50%.

Lines 16-31: Authors must remove all abbreviations because they are not allowed in the Simple Summary.

Lines 28-31: The conclusion of the Simple Summary should be rewritten because, in its current form, it practically repeats the description of the results. The conclusion should show the implications of the results without repeating them.

Abstract

Lines 33-34: Authors should define the meaning of “SCFAs” and “GIT” first, whichever is used in each section of the manuscript.

Line 34: Before describing the main findings obtained in the current study, the authors should describe the main materials and methods used. For example, the animal species used as experimental material should be specified. The physiological stage and age of the animals used should be specified. The treatments evaluated should be described in detail. The experimental design and number of replicates or experimental units within each treatment should be specified. The duration of the experimental phase should be specified.

Lines 34-58: The abbreviations CAT, SOD, GSH-Px, T-AOC, IL1β, TNFα, IL6, IL10, TNFα, and ZO1 were not defined. Therefore, they should be defined the first time they are used. This correction applies to all abbreviations used in each section of the manuscript.

Lines 34-55: The description of the results should be improved by adding the observed significance values ​​for each group of response parameters evaluated. Even if some effects do not have statistically significant differences, the observed significance value should be specified.

Introduction

Lines 73-76: Are there references to support these claims?

Lines 86-95: Please provide more details about the previous studies that evaluated SCFAs. For example, specifying the doses of SCFAs used in the previous studies will help readers better understand what is new in the current study.

Line 95: The authors should add background on the effects of SCFAs on parameters related to immune response, such as those evaluated in the current study.

Lines 96-102: These lines show general information about antioxidant mechanisms in living organisms. However, they are not related to the treatments evaluated in the current study. Therefore, the authors should remove these lines and replace them with specific information about the effects of SCFAs on antioxidant enzymes in goats. This information should contain details such as SCFA doses and experimental periods. It should also specifically indicate which antioxidant enzymes were enhanced. This background will help to better understand what is new in the current study regarding the antioxidant effects of SCFAs.

Lines 104-109: Are there references to support these claims?

Line 109: After correcting the introduction section, authors should add a clear hypothesis about the expected outcomes of their tested treatments.

Lines 110-118: All abbreviations should be removed, and only the full names of the response variables should be used. Also, authors should include only the groups of response variables to be assessed in the objective rather than the name of each response variable. For example, “antioxidant status” should be indicated instead of the name of each antioxidant enzyme.

Material and methods

Lines 144-146: The authors should specify the procedures used to slaughter the goats and take the samples. The time after slaughter, when the liver and intestinal tissue samples were taken, should be specified to verify that the integrity of the tissues was not altered.

Lines 147-154: Authors should specify the procedures and laboratory equipment used to measure antioxidant enzymes. They should also specify how many tissue replicates were used within each goat. This information is important to verify that the replicates were adequate to detect statistically significant effects.

Lines 156-163: The authors should specify the procedures used to slaughter the goats and take the samples. The time after slaughter, when the liver and intestinal tissue samples were taken, should be specified to verify that the integrity of the tissues was not altered. Authors should specify the procedures and laboratory equipment used to measure tight junction proteins and interleukins. They should also specify how many tissue replicates were used within each goat. This information is important to verify that the replicates were adequate to detect statistically significant effects.

Lines 165-167: The description of the statistical analyses used is too ambiguous. The authors should indicate which statistical tests were applied to the data to verify their normality before the statistical analyses. The experimental design used should also be specified. The authors should justify why they used Duncan's multiple comparison test.

Results

Lines 170-293: The authors' description of results needs improvement. In several cases, they state, “there were no differences between groups,” which is too ambiguous. The authors should specify which of the treatments evaluated was not different from the control. In addition, in the figure caption, the authors should define the abbreviations used in bar graphs, such as NC, SA, SP, SB, CAT, SOD, GSH-Px, T-AOC, IL‐1β, TNF‐α, IL‐6, IL‐10, and TNF‐α.

Lines 296-322: The description of correlations should be improved by adding the observed “r” values. This information will help readers understand the magnitude of the association between the different response variables.

Discussion

Lines 621-631: These lines of discussion should be removed because the authors did not evaluate essential oils. Therefore, it is not scientifically valid to compare the effects of SCFAs versus the effects of essential oils.

Lines 632-658: This discussion is correct. However, the authors should add the doses of SCFAs used by those other studies to understand better what accounts for the variation between those results and those observed in the current study.

Lines 659-721: These lines of discussion of interleukins and tight junction proteins are correct. Therefore, I have no additional suggestions along these lines.

Conclusions

Lines 723-727: These lines should be removed because they only show background information on SCFAs. No background information should be added to this conclusions section.

Lines 728-725: All abbreviations used in this section should be removed. The rest of the information added is correct.

Comments on the Quality of English Language

The quality of the English language requires some changes

Author Response

After reviewing the manuscript entitled “Gastric Infusion of Short Chain Fatty Acids Improves Healthvia Enhance Liver and Intestinal Immune Response and Antioxidant Capacity in Goat”, the following suggestions were made it. The manuscript contains interesting information on the in vivo effects of short-chain fatty acids on the antioxidant and immune capacity of goat liver and intestinal tissue. However, major corrections are required before the manuscript can be accepted for publication. The individual corrections are shown below:

Simple Summary

Point: Lines 16-24: The authors used an excessive amount of background information, which could confuse the reader. Please shorten these lines by at least 50%.

Respond: Thanks for your comment, we have corrected the summary 

Point: Lines 16-31: Authors must remove all abbreviations because they are not allowed in the Simple Summary.

Respond: Thanks for your comment, we have removed all abbreviations

Point: Lines 28-31: The conclusion of the Simple Summary should be rewritten because, in its current form, it practically repeats the description of the results. The conclusion should show the implications of the results without repeating them.

Respond: Thanks for your comment, we have corrected the summary 

 Abstract

Point: Lines 33-34: Authors should define the meaning of “SCFAs” and “GIT” first, whichever is used in each section of the manuscript.

Respond: Thanks for your comment, we have defined the meaning of “SCFAs” and “GIT” first

Point: Line 34: Before describing the main findings obtained in the current study, the authors should describe the main materials and methods used. For example, the animal species used as experimental material should be specified. The physiological stage and age of the animals used should be specified. The treatments evaluated should be described in detail. The experimental design and number of replicates or experimental units within each treatment should be specified. The duration of the experimental phase should be specified.

Response: Thanks for your comment, we have added more details in this section.

Point: Lines 34-58: The abbreviations CAT, SOD, GSH-Px, T-AOC, IL‐1β, TNF‐α, IL‐6, IL‐10, TNF‐α, and ZO‐1 were not defined. Therefore, they should be defined the first time they are used. This correction applies to all abbreviations used in each section of the manuscript.

Response: Thanks for your comment, we have defined all abbreviations

Point: Lines 34-55: The description of the results should be improved by adding the observed significance values for each group of response parameters evaluated. Even if some effects do not have statistically significant differences, the observed significance value should be specified.

Introduction

Point: Lines 73-76: Are there references to support these claims?

Response: Thanks for your comment, we have added the references.

Point: Lines 86-95: Please provide more details about the previous studies that evaluated SCFAs. For example, specifying the doses of SCFAs used in the previous studies will help readers better understand what is new in the current study.

Response: Thanks for your comment, we have added some information about doses

Point: Line 95: The authors should add background on the effects of SCFAs on parameters related to immune response, such as those evaluated in the current study.

Response: Thanks for your comment, we have added background effects of SCFAs on parameters related to immune response,

Point: Lines 96-102: These lines show general information about antioxidant mechanisms in living organisms. However, they are not related to the treatments evaluated in the current study. Therefore, the authors should remove these lines and replace them with specific information about the effects of SCFAs on antioxidant enzymes in goats. This information should contain details such as SCFA doses and experimental periods. It should also specifically indicate which antioxidant enzymes were enhanced. This background will help to better understand what is new in the current study regarding the antioxidant effects of SCFAs.

Response: Thanks for your comment, we have added background about the effects of SCFAs on antioxidant enzymes in goats

Point: Lines 104-109: Are there references to support these claims?

Response: Thanks for your comment, we have added the references.

Point: Line 109: After correcting the introduction section, authors should add a clear hypothesis about the expected outcomes of their tested treatments.

Respond: Thanks for your comment, we have added clear hypothesis

Point: Lines 110-118: All abbreviations should be removed, and only the full names of the response variables should be used. Also, authors should include only the groups of response variables to be assessed in the objective rather than the name of each response variable. For example, “antioxidant status” should be indicated instead of the name of each antioxidant enzyme.

Respond: Thanks for your comment, we have corrected this point.

Material and methods

Point: Lines 144-146: The authors should specify the procedures used to slaughter the goats and take the samples. The time after slaughter, when the liver and intestinal tissue samples were taken, should be specified to verify that the integrity of the tissues was not altered.

Response: Thanks for your comment, we have added more details in M&M (sampling collection part).

Point: Lines 147-154: Authors should specify the procedures and laboratory equipment used to measure antioxidant enzymes. They should also specify how many tissue replicates were used within each goat. This information is important to verify that the replicates were adequate to detect statistically significant effects.

Point: Lines 156-163: The authors should specify the procedures used to slaughter the goats and take the samples. The time after slaughter, when the liver and intestinal tissue samples were taken, should be specified to verify that the integrity of the tissues was not altered. Authors should specify the procedures and laboratory equipment used to measure tight junction proteins and interleukins. They should also specify how many tissue replicates were used within each goat. This information is important to verify that the replicates were adequate to detect statistically significant effects.

Respond: Thanks for your comment, we have added more details in M&M (sampling collection part).

Point: Lines 165-167: The description of the statistical analyses used is too ambiguous. The authors should indicate which statistical tests were applied to the data to verify their normality before the statistical analyses. The experimental design used should also be specified. The authors should justify why they used Duncan's multiple comparison test.

Respond: Thanks for your comment, we have corrected the statistical analysis description

Results

Point: Lines 170-293: The authors' description of results needs improvement. In several cases, they state, “there were no differences between groups,” which is too ambiguous. The authors should specify which of the treatments evaluated was not different from the control. In addition, in the figure caption, the authors should define the abbreviations used in bar graphs, such as NC, SA, SP, SB, CAT, SOD, GSH-Px, T-AOC, IL‐1β, TNF‐α, IL‐6, IL‐10, and TNF‐α.

Point: Lines 296-322: The description of correlations should be improved by adding the observed “r” values. This information will help readers understand the magnitude of the association between the different response variables.

Respond: Thanks for your comment, we have added “r” values.

Discussion

Point: Lines 621-631: These lines of discussion should be removed because the authors did not evaluate essential oils. Therefore, it is not scientifically valid to compare the effects of SCFAs versus the effects of essential oils.

Respond: Thanks for your comment, we have removed this paragraph.  

Point: Lines 632-658: This discussion is correct. However, the authors should add the doses of SCFAs used by those other studies to understand better what accounts for the variation between those results and those observed in the current study.

Point: Lines 659-721: These lines of discussion of interleukins and tight junction proteins are correct. Therefore, I have no additional suggestions along these lines.

Conclusions

Point: Lines 723-727: These lines should be removed because they only show background information on SCFAs. No background information should be added to this conclusions section.

Respond: Thanks for your comment, we have removed background information

Point: Lines 728-725: All abbreviations used in this section should be removed. The rest of the information added is correct.

Respond: Thanks for your comment, abbreviations have removed

Comments on the Quality of English Language

The quality of the English language requires some changes

Reviewer 4 Report

Comments and Suggestions for Authors

This research studied the impact of a blend of three short‐chain fatty acids (SCFAs): sodium acetate (SA), propionate (SP), and butyrate (SB) on the health and oxidative stress of goats. GIT health and innate immunity were assessed by studying the activity of antioxidant enzymes, such as Superoxide dismutase, Catalase, Maleic dialdehyde, Glutathione, and Glutathione peroxidase. Similarly, the total antioxidant capacity of GIT was examined by studying inflammatory cytokines and tight junction proteins. The results showed that using SCFAs enhanced the activity of some antioxidant enzymes, pro‐inflammatory and anti‐inflammatory cytokines, and tight junction proteins.

There are some issues that need to be addressed to improve the quality of the manuscript.

  1. Table 1's caption is incorrect. “Table 1. This is a table. Tables should be placed in the main text near to the first time they are cited.
  2. Goats Management and Experimental Design are unclear, and more details must be added.
  3. Line 140, D, dry matter, change D to DM
  4. In lines 144 and 156, “Hepatic and intestinal tissues were immediately extracted and homogenized...” It is not clear from where and how the samples were obtained.
  5. There are no letters above some bars to show the significant differences between the groups in all the figures, so please add them.
  6. The information on pages 14-22 is the same as on pages 5-13. Please delete them.

Author Response

Point: There are some issues that need to be addressed to improve the quality of the manuscript.

Table 1's caption is incorrect. “Table 1. This is a table. Tables should be placed in the main text near to the first time they are cited.”

Respond: Thanks for your comment, we have corrected this mistake

Point: Goats Management and Experimental Design are unclear, and more details must be added.

Respond: Thanks for your comment, we have modified M&M section

Point: Line 140, D, dry matter, change D to DM

Respond: Thanks for your comment, we have corrected this point

Point: In lines 144 and 156, “Hepatic and intestinal tissues were immediately extracted and homogenized...” It is not clear from where and how the samples were obtained.

Response: Thanks for your comment, we have added more details in M&M (sampling collection part)

Point: There are no letters above some bars to show the significant differences between the groups in all the figures, so please add them.

The information on pages 14-22 is the same as on pages 5-13. Please delete the

Round 2

Reviewer 1 Report

Comments and Suggestions for Authors

Authors have attended all observations that I suggested.

Thanks

Reviewer 3 Report

Comments and Suggestions for Authors

Comments and Suggestions for Authors

After reviewing the manuscript entitled “Gastric Infusion of Short Chain Fatty Acids Improves Healthvia Enhance Liver and Intestinal Immune Response and Antioxidant Capacity in Goat”, the following suggestions were made it. In this second (corrected) version of the manuscript, the authors incorporated most of my suggested corrections to the first version. The authors also provided appropriate responses to my comments. Consequently, I have no additional suggestions and believe the manuscript can be accepted for publication in its current form.

Round 3

Reviewer 1 Report

Comments and Suggestions for Authors

Authors have attended all observations that I recommended.

Thanks